

# Epidemiological and genetic characteristics of swine pseudorabies virus in mainland China between 2012 and 2017

Ying Sun[1,*], Wan Liang[1,2,*], Qingyun Liu[1,3], Tingting Zhao[1,3], Hechao Zhu[1,3], Lin Hua[1,3], Zhong Peng[1,3], Xibiao Tang[1], Charles W. Stratton[4], Danna Zhou[2], Yongxiang Tian[2], Huanchun Chen[1,3] and Bin Wu[1,3]

[1] The Cooperative Innovation Center for Sustainable Pig Production, College of Animal Science and Veterinary Medicine, Huazhong Agricultural University, Wuhan, China
[2] Key Laboratory of Prevention and Control Agents for Animal Bacteriosis (Ministry of Agriculture), Institute of Animal Husbandry and Veterinary Science, Hubei Academy of Agricultural Sciences, Wuhan, China
[3] State Key Laboratory of Agricultural Microbiology, Huazhong Agricultural University, Wuhan, China
[4] Department of Pathology, Microbiology and Immunology, Vanderbilt University Medical Center, Nashville, TN, United States of America
[*] These authors contributed equally to this work.

Corresponding authors
Zhong Peng,
pengzhong@mail.hzau.edu.cn,
pengzhong525@163.com
Bin Wu, wub@mail.hzau.edu.cn

## ABSTRACT

The outbreak of pseudorabies (PR) in many Bartha-K61 vaccinated farms in China in late 2011 has seriously damaged the pig industry of one of the largest producers of pork products in the world. To understand the epidemiological characteristics of the pseudorabies virus (PRV) strains currently prevalent in China, a total of 16,256 samples collected from pig farms suspected of PRV infection in 27 Provinces of China between 2012 and 2017 were evaluated for detection of PRV. Since the extensive use of gE-deleted PRV vaccine in China, the PRV-gE was applied for determining wild-type virus infection by PCR. Of the 16,256 samples detected, approximately 1,345 samples were positive for the detection of PRV-gE, yielding an average positive rate of 8.27%. The positive rates of PRV detection from 2012 to 2017 were 11.92% (153/1284), 12.19% (225/1846), 6.70% (169/2523), 11.10% (269/2424), 5.57% (147/2640), and 6.90% (382/5539), respectively. To understand the genetic characteristics of the PRV strains currently circulating, 25 PRV strains isolated from those PRV-gE positive samples were selected for further investigation. Phylogenetic analysis based on gB, gC, and gE showed that PRV strains prevalent in China had a remarkably distinct evolutionary relationship with PRVs from other countries, which might explain the observation that Bartha-K61 vaccine was unable to provide full protection against emergent strains. Sequence alignments identified many amino acid changes within the gB, gC, and gE proteins of the PRVs circulating in China after the outbreak compared to those from other countries or those prevalent in China before the outbreak; those changes also might affect the protective efficacy of previously used vaccines in China, as well as being associated in part with the increased virulence of the current PRV epidemic strains in China.

## INTRODUCTION

Pseudorabies virus (PRV) mainly causes reproductive failure in sows as well as respiratory and neurological symptoms in piglets (*Mettenleiter, 2000*; *Nauwynck et al., 2007*). PRV possesses a double-stranded liner DNA genome which contains more than 70 functional genes encoding proteins participating in the formation of viral capsid, tegument, and envelope (*Pomeranz, Reynolds & Hengartner, 2005*). Among these proteins, the envelope component proteins gB and gC induce cellular and humoral imMune responses (*Ober et al., 1998*; *Ober et al., 2000*), while gE acts as a major virulence determinant of PRV to pigs (*Kimman et al., 1992*; *Wang et al., 2014*). These three genes are commonly used for monitoring the evolution of PRV (*Muller et al., 2011*; *Sozzi et al., 2014*; *Yu et al., 2014*; *Wang et al., 2015*).

The first report of a PRV outbreak in China occurred in the 1950s. In the 1970s, an inactivated vaccine derived from PRV strain Bartha-K61 was imported from Hungary to China (*Yuan et al., 1983*; *An et al., 2013*). The widespread use of this vaccine in China was able to control outbreaks of pseudorabies between 1990 and 2011 (*Tong & Chen, 1999*). However, since late 2011, a pseudorabies (PR)-like disease has occurred in many Chinese pig farms that had been vaccinating pigs with the Bartha-K61 vaccine. PRV has been finally confirmed to be responsible for those outbreaks (*An et al., 2013*; *Peng et al., 2013*; *Luo et al., 2014*; *Wang et al., 2014*; *Yu et al., 2014*). A number of studies have noted that the Bartha-K61 vaccine appears to be unable to provide full protection against PRV strains isolated from those outbreaks (*An et al., 2013*; *Wang et al., 2014*; *Yu et al., 2014*). These findings suggest that there may be important changes in the PRVs currently circulating in China. However, genetic information as well as epidemiological data about PRV strains currently circulating in China is limited. Therefore, in this study, we report the detection/genetic analysis of PRVs recovered from pigs in China between 2012 and 2017. The aim of this study is to understand the epidemiological and genetic characteristics of PRVs that are currently prevalent in China.

## MATERIALS AND METHODS

### Samples collection and virus isolation

A total of 16,256 samples including tissue from lungs, lymph nodes, brains, serums, stillbirths, kidneys, and spleens were collected from pigs of different ages with signs suspected of PRV infection in farms (no. of sows ≥100) in 27 Provinces in mainland China, excluding Ningxia and Tibet, between January, 2012 and December, 2016 (Fig. 1). The number of samples collected in 2012, 2013, 2014, 2015, 2016 and 2017 was 1,284, 1,846, 2,523, 2,424, 2,640 and 5,539, respectively. After collection, tissues were minced, immersed with Dulbecco's modified Eagle medium (DMEM), and homogenized using a QIAGEN TissueLyser II (QIAGEN Germany). Sample homogenates were then frozen at −80 °C and thawed for three times. Following centrifugation at 5,000 rpm for 5 min, the supernatants were harvested for DNA and/or virus isolation.

Template DNA for PCR detecting PRV was isolated using a Universe Genomic DNA Kit (CWBIO, Beijing, China) following the manufacturer's instructions. For virus isolation,

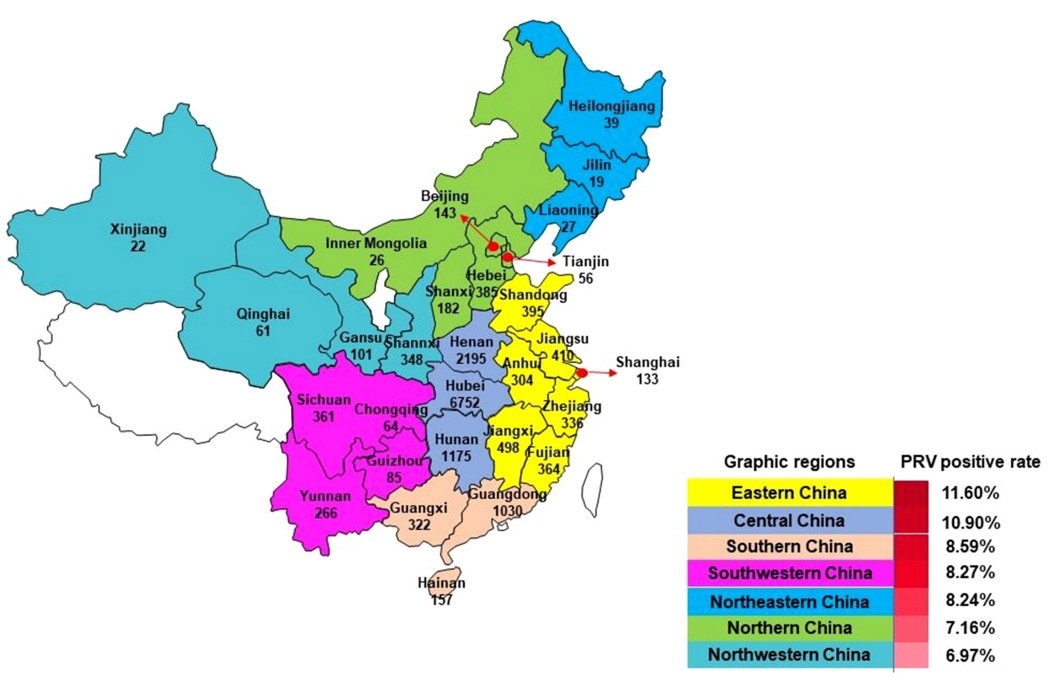

**Figure 1** Samples collection from mainland China for PRV detection between 2012 and 2017.

homogenate supernatants were filtered using a 0.22 μm membrane, and inoculated into PK-15 cells (Purchased from ATCC, Manassas, VA, USA). Cells were then incubated in a 37 °C incubator supplemented with 5% CO2. Cell culture with obvious CPE was used for further plaque purification assays. Briefly, PK-15 cells were plated into a 6-well plate, and a series of 10-fold dilutions (from $10^{-1}$ to $10^{-6}$) of virus was inoculated. The plate was incubated at 37 °C under an atmosphere containing 5% $CO_2$ for 2 h and was shaken one time every 15 min. After incubation, the virus was discarded and the cells were washed using DMEM for two times. Finally, DMEM medium containing 1% low melting agarose was added into each of the wells, and the plate was incubated at 4 °C until the medium solidified. The plate was then moved into a 37 °C cell incubator for plaque formation. Plaques with suitable size were selected and inoculated into 500 μL DMEM, frozen and thawed for three times, and then diluted 2-fold in DMEM for the second round of plaque purification assay. After that, plaque fluid was inoculated into PK-15 cells and cultured in flask.

## PRV detection

Polymerase chain reaction (PCR) assays were designed to detect the presence of PRV gE gene from the clinical samples using the DNA isolated as template as well as the primers listed in Table 1. The gE gene was used as the target gene because the gE-deleted pseudorabies virus (PRV) vaccine has been used in China extensively, and the detection of the gene could be applied for determining wild-type virus infection. As shown in Table 1, primers gE1-F and gE1-R were designed for the detection of gE. PCR reactions were performed in a 25 μL

**Table 1  Primers used in this study.**

| Primers | Sequences (5′–3′) | Products (bp) | Effects |
|---|---|---|---|
| gE1-F | CGTGTGGCTCTGCGTGCTGT | 342 | Sample detection |
| gE1-R | ATTCGTCACTTCCGGTTTC | | |
| gB2-F | GGCTGGTGGCGGTGTTTGGCG | 892 | Amplifying gB |
| gB2-R | AGGGCGAAGGAGTCGTAGGG | | |
| gC1-F | CCATGTGYGCCACTAGCATT | 965 | Amplifying the N-terminal of gC |
| gC1-R | CGGTGCTGTTGGTCACGAAG | | |
| gC2-F | CAACGTCTCGCTCCTCCTGT | 921 | Amplifying the C-terminal of gC |
| gC2-R | GCCGTCGTCTCGTGTGGTT | | |
| gE2-F | GACCATGCGGCCCTTTCTGC | 899 | Amplifying the N-terminal of gE |
| gE2-R | GGTCCACCGGGCGCAGGCA | | |
| gE3-F | TTTACCGCCACGCTGGACTGGT | 1,098 | Amplifying the C-terminal of gE |
| gE3-R | CTTGGGGGCCAGCAGGACGT | | |

volume mixture containing 12.5 μL 2×Taq Master mix (TAKARA, Shiga, Japan), 8.5 μL nucleotide-free water, each of the forward and reverse primers 1 μL, 1 μL DMSO, and 1 μL template DNA. Thermocycler conditions used for PCR were 95 °C for 5 min, followed by 35 cycles of denaturation at 95 °C for 30 s, annealing at 55 °C, 30 s for gE, and extension at 72 °C for 1 min, with a final extension at 72 °C for 10 min before storage at 4 °C. The PCR product was visualized using 1% agarose gel electrophoresis under ultraviolet light.

## Sequencing and phylogenetic analysis

PCR assays were also designed for analysing the gB, gC, and gE genes of the PRVs currently circulating in China. The PCR volumes were the same as that used for PRV detection from the samples. Cycling conditions were 95 °C for 5 min, followed by 35 cycles of denaturation at 95 °C for 30 s, annealing at 60 °C for 30 s, and extension at 72 °C for 1 min, with a final extension at 72 °C for 10 min before storage at 4 °C. The PCR product was visualized using 1% agarose gel electrophoresis under ultraviolet light.

After amplification, PCR products were purified using a TIANgel Midi Purification Kit (Tiangen Biotechnique Inc., Beijing, China) and cloned into a pMD19-T vector (TAKARA, Shiga, Japan). Plasmids carrying either gB, gC, or gE were extracted and sent to Genscript (Nanjing, China) for DNA sequencing. The nucleotide sequences of gB, gC, and/or gE were edited and aligned (using BioEdit software) and then translated into aminoacids from each gene and then compared using DNAStar. Phylogenetic trees were generated through MEGA X, using neighbor-joining algorithm with 1,000 bootstrapping. Sequences of PRV strains listed in Table 2 retrieved from NCBI were used as references.

## RESULTS

### PCR detection of PRV in mainland China

Of the 16,256 samples, 1,345 samples were positive for the detection of PRV-gE, yielding an average positive rate of 8.27%. The positive rates of PRV detection from 2012 to 2017 were 11.92% (153/1284), 12.19% (225/1846), 6.70% (169/2523), 11.10% (269/2424),

**Table 2** PRV reference strains used in this study.

| Strain | Year of isolation | Place of isolation | GenBank accession |
|---|---|---|---|
| Bartha | — | Hungary | JF797217.1 (complete genome) |
| Becker | — | United States | JF797219.1 (complete genome) |
| Kaplan | — | Hungary | JF797218.1 (complete genome) |
| NIA3 | — | Japan | KU900059.1 (complete genome) |
| Fa | 2001 | China | KM189913.1 (complete genome) |
| TJ | 2012 | China/Tianjin | KJ789182.1 (complete genome) |
| BJ/YT | 2012 | China/Beijing | KC981239.1 (complete genome) |
| ZJ01 | 2012 | China/Zhejiang | KM061380.1 (complete genome) |
| HN1201 | 2012 | China/Henan | KP722022.1 (complete genome) |
| HNX | 2012 | China/Henan | KM189912.1 (complete genome) |
| HeN1 | 2012 | China/Henan | KP098534.1 (complete genome) |
| HLJ8 | 2013 | China/Heilongjiang | KT824771.1 (complete genome) |
| Ea | 1999 | China/Hubei | AF257079.1 (gB), AF158090.1 (gC), AF171937.1 (gE) |
| GX-NL | 2007 | China/Guangxi | KT948044.1 (gB), KU323908.1 (gC), KT936469.1 (gE) |
| GD-SH | 2007 | China/Guangdong | KT948054.1 (gB), KU323907.1 (gC), EF552427.1 (gE) |
| GD-GZ | 2009 | China/Guangdong | KT948042.1 (gB), KU323905.1 (gC), KT936466.1 (gE) |
| GD-GZ2 | 2013 | China/Guangdong | KT948045.1 (gB), KU323903.1 (gC), KT936467.1 (gE) |
| HN-CZ | 2013 | China/Hunan | KT948049.1 (gB), KU323912.1 (gC), KT936465.1 (gE) |
| GD-FS | 2014 | China/Guangdong | KT948040.1 (gB), KU323909.1 (gC), KT936476.1 (gE) |
| GD-HS2 | 2014 | China/Guangdong | KT948047.1 (gB), KU323911.1 (gC), KJ660063.1 (gE) |
| GD-JM | 2015 | China/Guangdong | KT948048.1 (gB), KU323899.1 (gC), KT936473.1 (gE) |
| GD-QY | 2010 | China/Guangdong | KT948053.1 (gB), KU323901.1 (gC) |
| GX-GL | 2013 | China/Guangdong | KT948046.1 (gB), KU323910.1 (gC) |
| GD-YF | 2015 | China/Guangdong | KT948041.1 (gB), KU323904.1 (gC) |
| P-PrV | 2003 | Malaysia | EU915280.1 (gC), FJ176390.1 (gE) |
| LXB6 | 2009 | China/Heilongjiang | GQ926931.1 (gC), GQ926932.1 (gE) |
| SMX | 2014 | China/Henan | KR025920.1 (gC), KP192495.1 (gE) |
| GD-WH | 2015 | China/Guangdong | KU323902.1 (gC), KT936468.1 (gE) |
| HNXX | 2012 | China/Henan | KJ526436.1 (gB), KJ526441.1 (gC) |
| HS | 2008 | China/Sichuan | EU719636.1 (gC) |
| Min-A | 2002 | China/Fujian | AY170318.1 (gE) |

5.57% (147/2640), and 6.90% (382/5539), respectively. Monthly, higher positivity rates of PRV were detected in January, February, March, April, June, October, November and December; and winter (December, January and February), spring (March, April, and May) and autumn (September, October and November) were the seasons with the high positivity rate of PRV detection during 2012 and 2017 (Figs. 2A and 2B).

Mainland China is divided into seven parts including Northeastern China, Northern China, Eastern China, Central China, Southern China, Northwestern China, and Southwestern China (Fig. 1). Among these graphic regions, the positive rates of PRV detection in Eastern China and Central China between 2012 and 2017 were higher than 10.00%, while the positive rates in other parts of China between 2012 and 2017 ranged

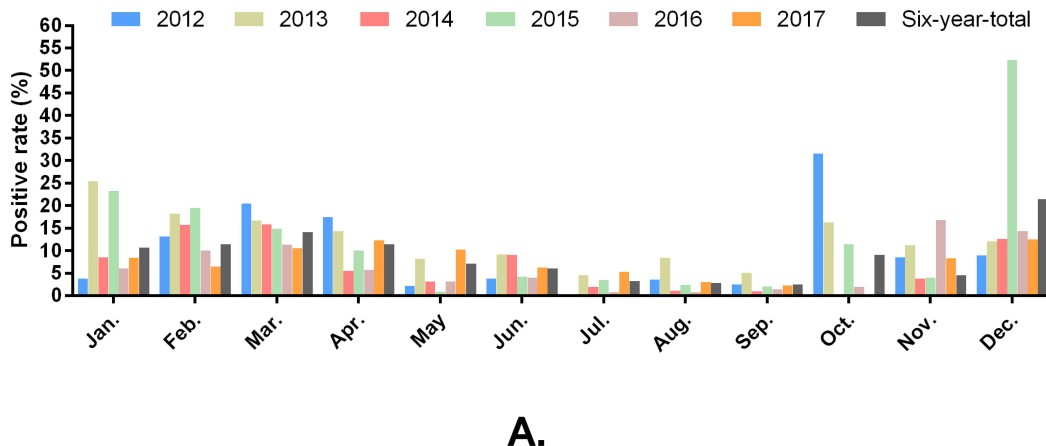

**A.**

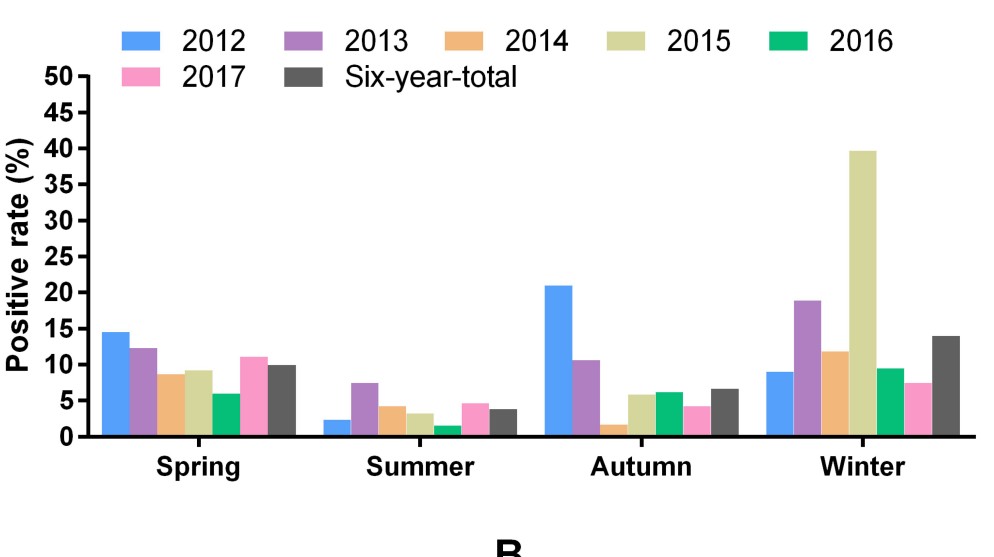

**B.**

**Figure 2** **Positivity rate of PRV detection in different months (A) and seasons (B).**

from approximately 7.00% and 10.00% (Fig. 1). In addition, the positivity rate of PRV detection in different graphic regions between different years displayed diversity. For instance, the positive rates of PRV detection in Northern China in 2012, 2013 and 2015 were higher than 15.00%, but the positivity rates in 2014 and 2016 were lower than 5.00%; the positive rate of PRV detection in Southwestern China in 2012 was 26.15%, while it was only 1.80% in 2013, and 6.90% in 2014, 5.97% in 2015, but zero in 2016; in Northwestern China, the positivity rate of PRV detection in 2012 and 2013 were higher than 20.00%, and approximately 10.40% in 2014 and 2015, but zero in 2016 (Fig. 3).

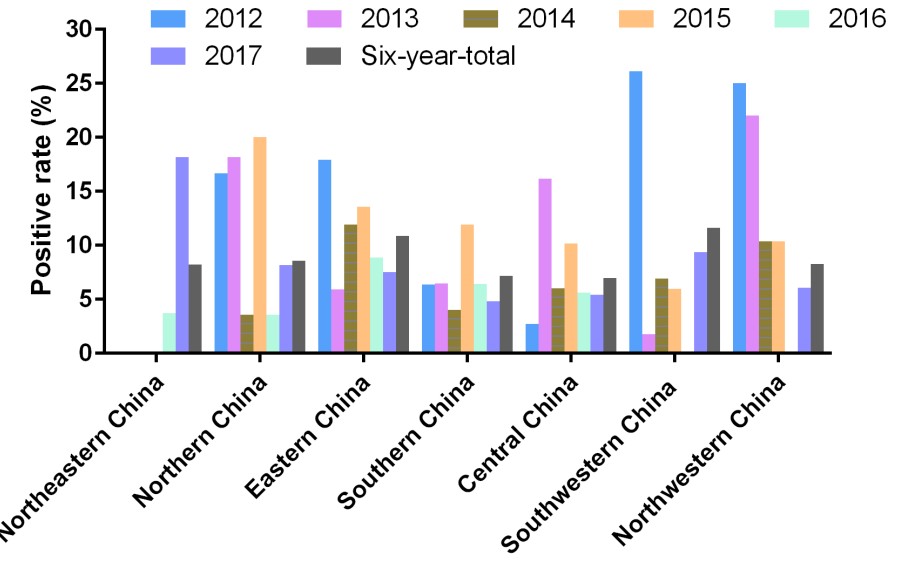

**Figure 3  Positivity rate of PRV detection in different graphic parts of China.**

## PRV isolation

To understand the genetic characteristics of PRVs currently circulating in China, a total of 25 PRV strains isolated herein were used for further analysis in this study (Table 3). Most of those strains were isolated from lungs and their $TCID_{50}$/0.1 mL values ranged from $10^{6.72}$ to $10^{7.96}$.

## Phylogenetic characteristics

Phylogenetic analysis using gB, gC, and/or gE sequence showed that the PRV isolates from China were located on a phylogenetic branch, which was distinct from the those isolates from other countries of the world (Figs. 4A, 4B, 4C). According to previous study (*Ye et al., 2015*), those isolates from China belonged to genotype II, while those isolates from the other parts of the world including Bartha, Becker, Kaplan, and NIA3 were genotype I strains. Interestingly, one isolate from China in 2016, which we designed HuB, should be determined as a genotype II strain (Fig. 4B) according to Ye's study, in which PRVs were phylogenetically divided into two major genotypes, genotype I and genotype II, based on the analysis of gC gene (*Ye et al., 2015*). However, this isolate had a closer relationship to the genotype I strains when using gB to perform the phylogenetic analysis (Fig. 4A).

## Analysis of gB, gC and gE

The maximal amino acid sequence divergence for gB, gC, and gE proteins of the 25 PRV isolates were 5.2, 2.7, and 2.6% within the isolates, and were 8.4, 9.2, and 5.5% compared to those isolates from the other countries, respectively. The maximal amino acid sequence divergence for the three proteins of the 25 isolates were 4.8, 9.9, and 2.8% compared to those strains prevalent in China before 2012, and were 4.8, 2.7, and 2.8% after 2012.

Alignment of amino acid sequences of gB found the isolates from China mainly had three types of mutations within the protein compared those strains from the other countries. The

**Table 3    Twenty-five strains of PRV isolated and analyzed in this study.**

| Strains | Place of isolation | Samples of isolation | Date of isolation | TCID50/0.1 mL |
|---------|--------------------|-----------------------|--------------------|----------------|
| HeNFJC | Henan | Lung | 2015/11 | $10^{6.72}$ |
| HeNJYG | Henan | Lung | 2015/11 | $10^{7.38}$ |
| GDFC | Guangdong | Lung | 2016/3 | $10^{7.00}$ |
| GD | Guangdong | Lung | 2016/3 | $10^{7.25}$ |
| HeNZZZM | Henan | Lung | 2016/3 | $10^{7.25}$ |
| HuBLLP | Hubei | Lung | 2016/3 | $10^{7.59}$ |
| FJFQ | Fujian | Lung | 2016/3 | $10^{7.28}$ |
| GDHDYC | Guangdong | Lung | 2016/3 | $10^{7.49}$ |
| HuN | Henan | Brain | 2016/3 | $10^{7.96}$ |
| SDRZ | Shandong | Lymph nodes | 2016/3 | $10^{7.36}$ |
| HuBYCYJ | Hubei | Lymph nodes | 2016/4 | $10^{7.67}$ |
| HeNXY | Henan | Lung | 2016/4 | $10^{7.08}$ |
| HuBWX | Hubei | Lung | 2016/4 | $10^{7.25}$ |
| SDSCL | Shandong | Lung | 2016/5 | $10^{7.80}$ |
| HuBHC | Hubei | Lung | 2016/5 | $10^{7.59}$ |
| HuBAL | Hubei | Brain | 2016/9 | $10^{7.12}$ |
| HuBZX | Hubei | Brain | 2016/9 | $10^{7.40}$ |
| ZJHY | Zhejiang | Lung | 2016/10 | $10^{7.25}$ |
| SX | Shanxi | Lung | 2016/10 | $10^{7.57}$ |
| HeNXP | Henan | Brain | 2016/11 | $10^{7.25}$ |
| HuBHZ | Hubei | Lung | 2016/11 | $10^{7.12}$ |
| HuBQJ | Hubei | Lung | 2016/11 | $10^{7.00}$ |
| HuB | Hubei | Tonsil | 2016/12 | $10^{7.35}$ |
| JSZL | Jiangsu | Brain | 2016/12 | $10^{7.43}$ |
| HuBSP | Hubei | Brain | 2016/12 | $10^{7.54}$ |

isolates from China harbored an-amino acid insertion at site 94 (G), a substitution of ten amino acids at sites 53 (A→T), 55 (P→T), 70 (T→A), 81 (N→D), 82 (D→G), 83 (V→F), 87 (A→E), 93 (E→D), 96 (F→V) and 102 (E→D), and a deletion of three-amino acids at sites 75–77 (S, P and G) compared to Bartha-Hungary, Kaplan-Hungary and NIA-3-Japan. In addition, there were also some different substitutions at different sites within the gB protein of the 25 isolates compared to Bartha-Hungary. For example, HuBHZ-China-2016 harbored a substitution of one amino acid at sites 11 (P→A) and 12 (R→G), while strains JSZL-China-2016, HuB-China-2016, and SDRZ-China-2016 had an amino acid substitution at sites 12 (R→H), 67 (A→V), and 228 (K→E), respectively (Fig. 5).

For the gC protein, the isolates from China had an insertion of seven amino acids at sites 63-69 (A, A, A, S, T, P and A) within the protein compared to the genotype I strains; while strain LXB6-China-2009 harbored an insertion of six amino acids at sites 64-69 (A, A, S, T, P and A). In particular, those isolates from China after 2012 contained a substitution of twenty-three amino acids substitutions at sites 14 (P→L), 16 (A→T), 52 (P→S), 55 (A→E), 57 (A→V), 59 (P→G), 60 (E →T), 76 (A→V), 87 (P→Q), 90 (N→G), 102 (A→S), 130 (F→V), 163 (S→P), 186 (T→A), 188 (V→A), 190 (E→V), 191 (D →V), 243
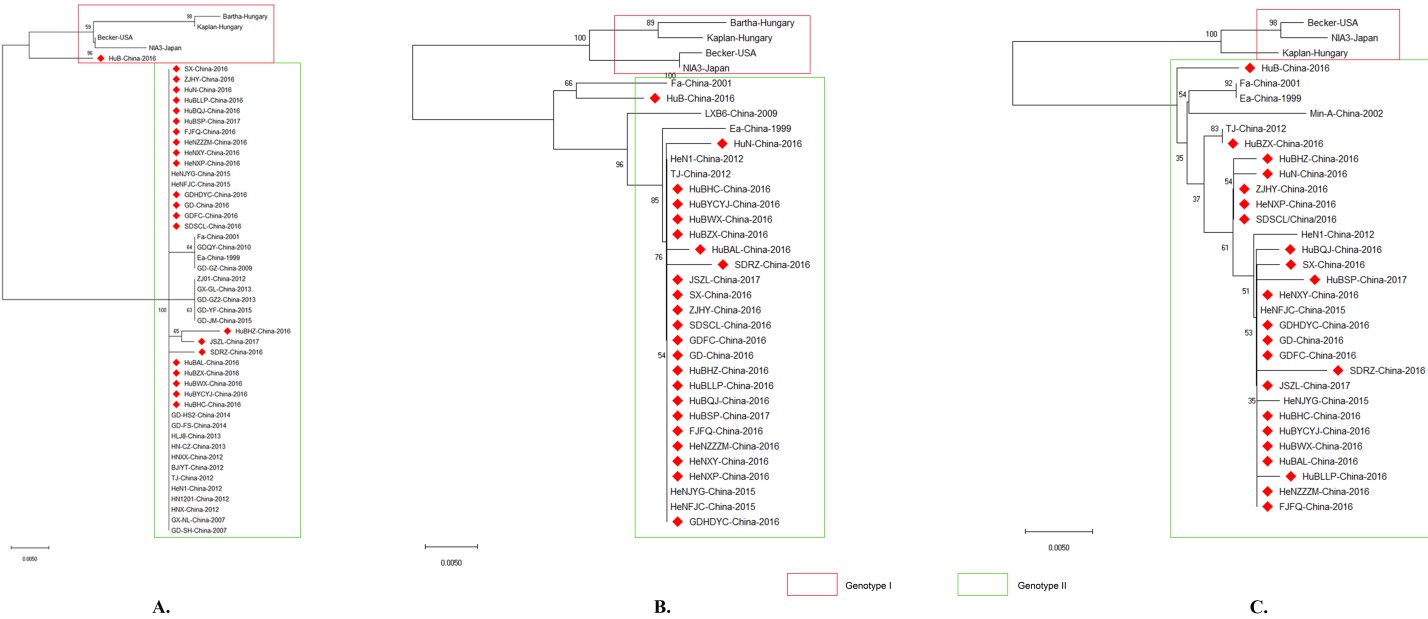

**Figure 4** **Evolutionary relationships of PRV isolates based on gB (A), gC (B) and gE (C).** The evolutionary history was inferred using the Neighbor-Joining method [1]. The optimal tree with the sum of branch length = 0.08532423 (gB)/ 0.11049107 (gC)/ 0.08878838 (gE) is shown. The percentage of replicate trees in which the associated taxa clustered together in the bootstrap test (1,000 replicates) are shown next to the branches. The tree is drawn to scale, with branch lengths in the same units as those of the evolutionary distances used to infer the phylogenetic tree. The evolutionary distances were computed using the p-distance method and are in the units of the number of amino acid differences per site. The analysis involved 50/34/33 amino acid sequences. All positions containing gaps and missing data were eliminated. There were a total of 293/462/570 positions in the final dataset. Evolutionary analyses were conducted in MEGA X.

(S→H), 431 (L→M), 449 (A→T), 457 (S→T), 461 (V →T) and 467 (G→A) compared to the genotype I strains (Fig. 6). In addition, some other substitutions were also found during the analysis: HuN-China-2016, SDRZ-China-2016, HuB-China-2016, HuN-China-2016, HuBAL-China-2016, and SDRZ-China-2016 harbored amino acid substitutions at sites 106 (K→T), 107 (R→C), 210/227 (A→T), 235 (A→V), 300 (L→P) and 386 (W →R), respectively.

The sequence alignments of the gE protein found also found some amino acid mutations within the protein of the genotype II strains compared to the genotype I isolates. Compared with Kaplan-Hungary and NIA3-Japan, the 25 isolates contained an insertion of one amino acid (D) at site 48 (Fig. 7). Compared with Min-A-China-2002, the 25 isolates contained a deletion of one amino acid (D) at site 493. In particular, HeNJYG-China-2016, HuBWX-China-2016, HeNXY-China-2016 and GDHDYC-China-2016 deleted an amino acid (D) at site 491, while HuN-China-2016 and SDRZ-China-2016 had a deletion of one amino acid at sites 489 (Y) and 495 (D) compared to Min-A-China-2002, respectively; strain HuB-China-2016 deleted four amino acids (DLNG) at sites 61-64 compared to Min-A-China-2002. In addition to amino acid deletion, sequence alignments also identified amino acid substitutions within the gE protein of the strains isolated herein. For example, strains HeNJYG-China-2016, HuBHZ-China-2016, HuBLLP-China-2016, HuN-China-2016, HuN-China-2016, and SX-China-2016 contained one amino acid substitution at sites 336
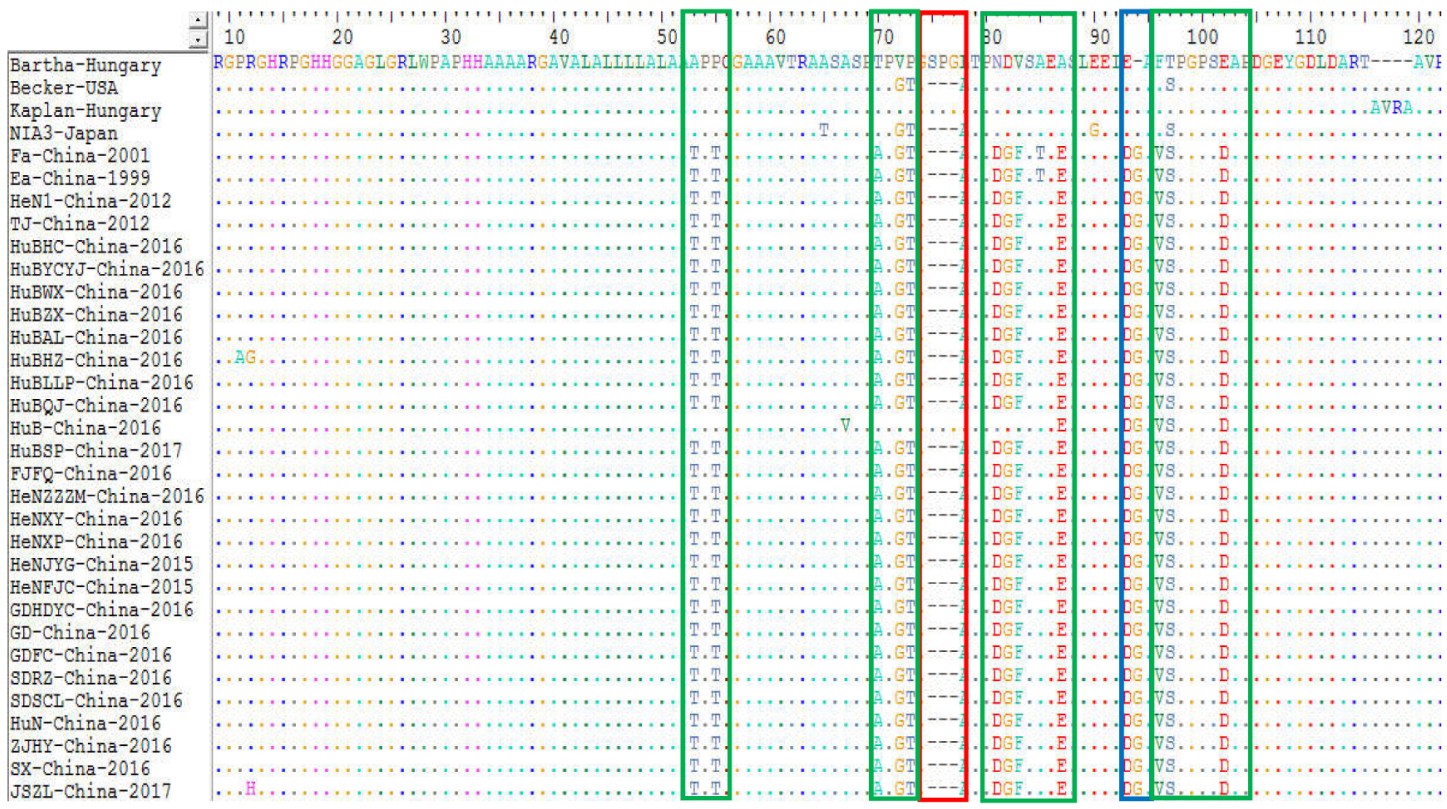

**Figure 5  Alignment of partial amino acid sequences of PRV gB protein.** The substitution regions are shown by the green boxes. The deletion region is shown by the red box. The insertion region is shown by the blue box.

(D→G), 2 (R→G), 473 (T→M), 49 (L→R), 573 (A→T), and 526 (D→G) compared to the genotype I strains, respectively. Particularly, one strain, SDRZ-China-2016, contained four substitutions at sites 407 (V→L), 487 (E→D), 499 (E→D) and 535 (E→D) compared to the genotype I strains.

## DISCUSSION

PRV is a common threat to the pig industry worldwide and is particularly important in China. The outbreak of PR in China in late 2011 has seriously damaged the pig industry of one of the largest producers of pork products in the world (*An et al., 2013*; *Yu et al., 2014*). The present study reported the prevalence of PRV in China between 2012 and 2017. This report is the first large-scale etiological investigation of PRV involved in most regions of China following the outbreak. The data revealed an average positive rate of 8.27% for PRV detection during the six years, and higher than 6.9% positivity rate of PRV detection in different regions in China (Fig. 1). While there is a lack of similar data from the other studies, a nationwide surveillance detecting the PRV-gE antibody revealed that the positive rate of PRV-gE antibody in China during 2013–2016 was higher than 13.74% (*Liu et al., 2018*). These findings confirm that the prevalence of PRV remains a problem in China.

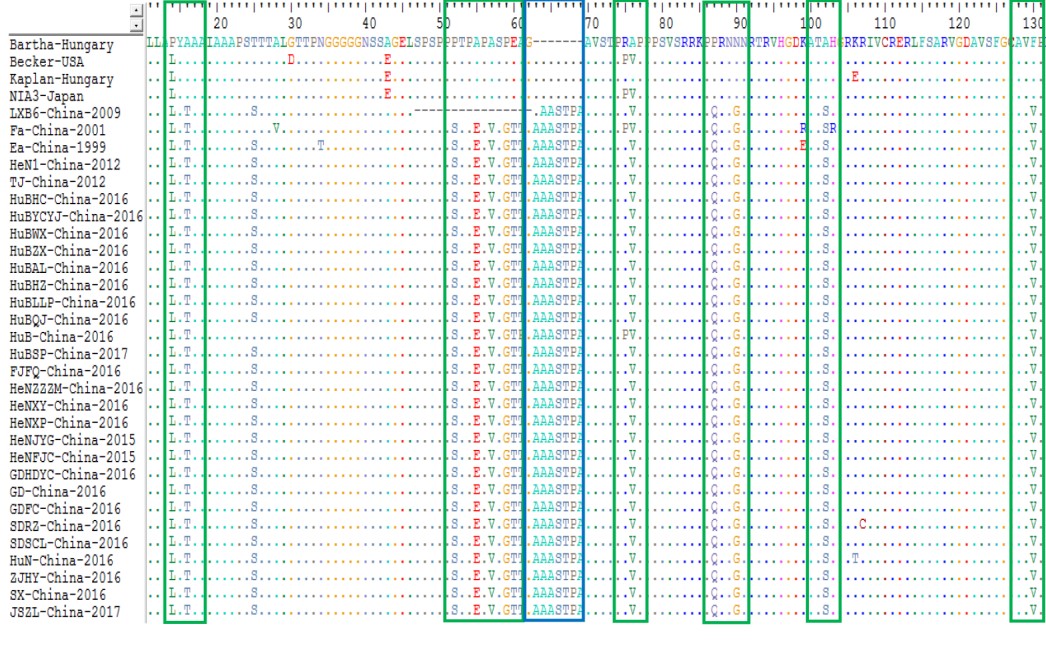

**Figure 6  Alignment of complete amino acid sequences of PRV gC protein.** The substitution regions are shown by the green boxes. The insertion region is shown by the blue box.

From the 1990s until late 2011, >80% of pigs in China were vaccinated with the Bartha-K61 vaccine, and pseudorabies was well controlled (*Yu et al., 2014*). This could be reflected by the unpublished data from Huazhong Agricultural University Diagnostic Center for Animal Infectious Diseases (Wuhan, China) which indicated that the detective

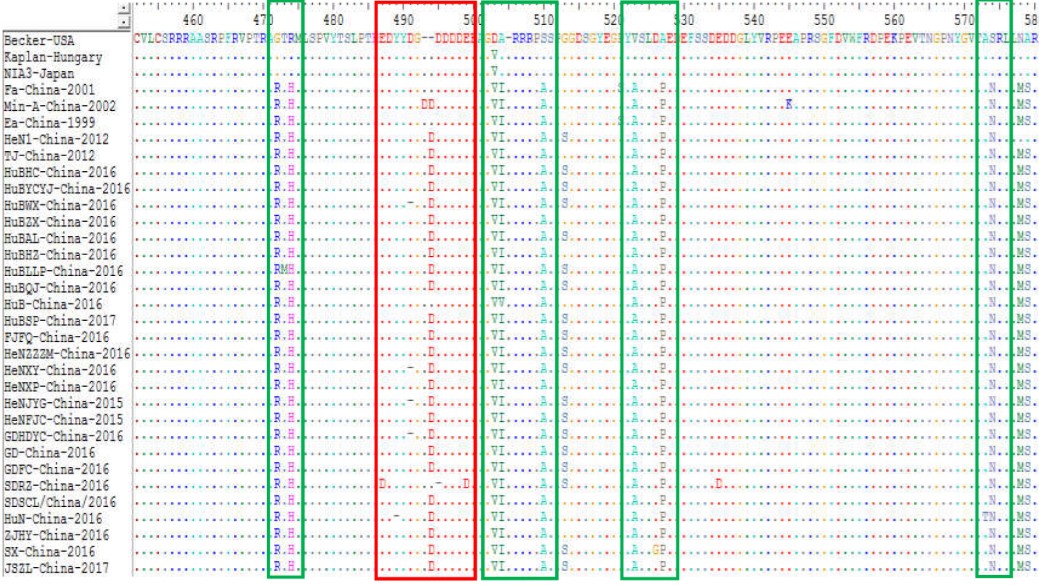

**Figure 7  Alignment of complete amino acid sequences of PRV gE protein.** The substitution regions are shown by the green boxes. The insertion region is shown by the blue box. The deletion regions are shown by the red boxes.

rate of PRV in mainland China between 2010 and 2011 was lower than 6%. However, the positive rate of PRV increased to approximately 12% between 2012 and 2013 as revealed by this study. These findings suggest that receiving the Bartha-K61 vaccine is likely to be unable to provide full protection against the PRVs circulating in China after 2011. Actually, this suggestion has been confirmed in laboratory (*An et al., 2013*; *Yu et al., 2014*).

The results from this study showed that the positive rate of PRV detection had a significant decreasing after 2013, this might be because several vaccines based on the epidemic strains have been developed after the 2011 outbreak, and the use of these vaccines helps to decrease the infection of the virus (*Wang et al., 2014*; *Hu et al., 2015*).

It has been reported that PRV isolates are generally divided into two genotypes according to the gC gene; PRV strains prevalent in China belong to genotype II while PRV isolates from the other countries belong to genotype I (*Ye et al., 2015*). In agreement with this report, the isolates analyzed in this study were divided into two distinct clusters according to their gC genes, with the previously reported genotype I strains Bartha, Backer, Kaplan and NIA3 forming one cluster and the Chinese strains including the reported genotype II strains TJ, JS, and HeN1 forming the second cluster (Fig. 4B). Interestingly, similar results were also obtained when using gB and gE to perform the phylogenetic analysis (Figs. 4A and 4C). Those findings again confirm that PRV strains circulating in China harbor different genotypes from those spreading in the other countries. This observation also might be the reason that the Bartha-K61 vaccine was unable to provide full protection against these emergent strains. Interestingly, one isolate, HuB-China-2016, belonged to genotype II according to gC and gE, but was identified as a genotype I strain when using gB as a phylogenetic criterion (Fig. 4). These findings suggest that a genetic recombination might have occurred within the genome of this isolate. In the next step of this investigation, we intend to do a follow up study in which the whole genome will be sequenced in order to clarify what happened with this strain.

The gB protein is the most conserved glycoprotein in herpesviruses and allows PRV strains to enter the target cells, thus contributing directly to cell-to-cell spread (*Mettenleiter, 2003*). In addition, this gB protein is the principle immunogen of the virus, stimulating the host to produce both complement-dependent and non-complement dependent neutralizing antibodies (*Okazaki, 2007*). Alignments of the gB protein indicated that the Chinese strains had amino acid insertions, deletions, and substitutions in comparison with strain Bartha-k61 (Fig. 5). These amino acid changes might lead to the alteration of the neutralizing epitope of the gB protein and thus alter the protective efficacy of previously used Bartha-k61vaccines in China.

The gC protein is another important neutralizing antigen and is the major virulent protein of PRV, guiding the adsorption process between the virus and target cells (*Karger, Schmidt & Mettenleiter, 1998*). Sequence alignments of the gC protein found that the most Chinese strains contained a continuous insertion of seven amino acids (AAASTPA at sites 63-69) and two-amino acid substitution within the protein compared to Bartha-k61 (Fig. 6). These changes might influence the structure of the gC glycoprotein of those strains, and therefore influence the virus adhesion to host cells. The gE protein is another major virulent protein of PRV (*Wang et al., 2015*). It has been reported that only a few amino acids changes are required to alter the virulence of PRV isolates (*Mettenleiter et al., 1994*). In this study, we found that the 25 PRV strains contained amino acid-insertion/deletion within the gE protein either compared to the isolates from other countries (Kaplan-Hungary and

NIA3-Japan) or compared to the isolates from China before 2012 (strain Min-A-China-2002); those changes might also have an effect on the virulence of PRV isolates currently circulating in China.

## CONCLUSION

In summary, this study reported a large-scale etiological investigation of PRV involved in most regions of China after the outbreak of PR in late 2011. Our data revealed an average positive rate of 8.27% for PRV detection between 2012 and 2017, indicating the risk of pseudorabies prevalence in China. Phylogenetic analysis showed that the evolutionary relationship between the PRV isolates circulating in China and those from the other countries was remarkably distinct, suggesting that vaccination with foreign strains might be unable to provide full protection against currently epidemic isolates in China. In addition, PRV isolates currently circulating contained different types of mutations within their gB, gC, and gE proteins compared to those from other countries and/or those from China before the outbreak; these changes also might be associated with virulence changes of the virus. In the next step, we intend to evaluate the influence of those changes on the virulence/pathogenicity of the isolates.

### Funding
This work was supported by the National Key R&D Program of China (Grant number: 2018YFD0500800), and the National Key Technology Support Program of China (Grant number: 2015BAD12B04). There was no additional external funding received for this study. The funders had no role in study design, data collection and analysis, decision to publish, or preparation of the manuscript.

### Grant Disclosures
The following grant information was disclosed by the authors:
National Key R&D Program of China: 2018YFD0500800.
National Key Technology Support Program of China: 2015BAD12B04.

### Competing Interests
The authors declare there are no competing interests.

### Author Contributions

- Ying Sun, Qingyun Liu and Tingting Zhao performed the experiments, prepared figures and/or tables, approved the final draft.
- Wan Liang performed the experiments, analyzed the data, contributed reagents/materials/analysis tools, prepared figures and/or tables, authored or reviewed drafts of the paper, approved the final draft.
- Hechao Zhu and Lin Hua performed the experiments, approved the final draft.

- Zhong Peng conceived and designed the experiments, analyzed the data, contributed reagents/materials/analysis tools, prepared figures and/or tables, authored or reviewed drafts of the paper, approved the final draft.
- Xibiao Tang conceived and designed the experiments, approved the final draft.
- Charles W. Stratton and Yongxiang Tian authored or reviewed drafts of the paper, approved the final draft.
- Danna Zhou analyzed the data, contributed reagents/materials/analysis tools, authored or reviewed drafts of the paper, approved the final draft.
- Huanchun Chen conceived and designed the experiments, authored or reviewed drafts of the paper, approved the final draft.
- Bin Wu conceived and designed the experiments, analyzed the data, authored or reviewed drafts of the paper, approved the final draft.

## Data Availability

NCBI GenBank with a BioProject ID: PRJNA250865 sequences are also available in the Supplemental File.

## Supplemental Information

Supplemental information for this article can be found online at http://dx.doi.org/10.7717/peerj.5785#supplemental-information.

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
