# Peer review of "Epidemiological and genetic characteristics of swine pseudorabies virus in mainland China between 2012 and 2017"

_PeerJ, doi:10.7717/peerj.5785_

## Round 0.1 · original submission · Major Revisions

Please read the reviewer comments one by one,and revise your manuscript. Especially, revise it according to Reviewer 2 comments.

·

Basic reporting

The paper is well written. It is clear and easy to read. It is an extensive epidemiological work based on PRV strains isolated in China. The background is relevant and the references are appropiated. Figures and tables are relevant

Experimental design

Methodology is described with sufficient detail to replicate by others. The analysis of the strains contributes to reevaluate the use of those types of vaccines in China

Validity of the findings

Data obtained were robust and very important in the epidemiology of PRV in China. References are well used througout the paper

Additional comments

I strongly recommend the publication of the work with minor changes. Please see a pdf attached to this review

Reviewer 2 ·

Basic reporting

no comment

Experimental design

The PRV was outbreak in late 2011. Epidemiological survey that cover the period before and after 2011 and compare the situation of clinical infection and the genetic diversity of PRV before and after 2011 will be more meaningful.

Validity of the findings

no comment

Additional comments

The manuscript investigated the prevalence of PRV from 2012 to 2017 in 27 provinces of China. A total of 16256 samples collected from large-scale farms suspected of PRV infection, and the positive rates of PRV, 5.57% to 12.19% for 2012-2017, were given. Some positive samples were DNA sequenced and aligned. The study has a large sample size and introduces the situation of clinical infection in detail.
> The PRV was outbreak in late 2011. Epidemiological survey that cover the period before and after 2011 and compare the situation of clinical infection and the genetic diversity of PRV before and after 2011 will be more meaningful.
> How about the situation of small farms in the backyard? Since the backyard farms usually don’t carry out good vaccination, when compared with large-scale farms.
> There is no further discussion and analysis on the results of PRV positive detection in different regions in different years.
> In the Abstract, please clearly give the method for determining wild-type virus. PCR or ELISA?
>“Of the 16256 samples detected, approximately 1345 samples were positive for the detection of PRV-gE,”the word " approximately " need to be removed.
>“To understand the genetic characteristics of PRVs currently circulating in China, a total of 25 PRV strains isolated herein were used for further analysis in this study” Why did the authors choose these 25 isolates? Are they representative?

---

## Round 0.2 · accepted · Accept

Thank you for your response and revision point-to-point according to the reviewers’ comments.

# ·

Basic reporting

no comments

Experimental design

no comments

Validity of the findings

no comments

Additional comments

Thank you for sending revised version. It is acceptable in the present form

Reviewer 2 ·

Basic reporting

no comment

Experimental design

no comment

Validity of the findings

no comment

Additional comments

There is no further question for the manuscript